# Association between Self-Reported Sleep Duration and Dietary Nutrients in Korean Adolescents: A Population-Based Study

**DOI:** 10.3390/children7110221

**Published:** 2020-11-08

**Authors:** Jee Hyun Lee, Sang-Jin Chung, Won Hee Seo

**Affiliations:** 1Department of Pediatrics, Korea University Ansan Hospital, Korea University College of Medicine, Ansan 15459, Korea; izzihn@gmail.com; 2Department of Foods and Nutrition, Kookmin University, Seoul 02707, Korea; schung@kookmin.ac.kr

**Keywords:** adolescent, nutrients, obesity, sleep duration

## Abstract

(1) Background: Adolescence is a transient period from childhood to adulthood, which is characterized by rapid physical growth and psychological changes, including sleep. Because the relationship between insufficient sleep and obesity has been observed in children and adults, the potential links between sleep, dietary intake, and nutrition have received increased attention. We aimed to examine the association of sleep duration with dietary nutrients intake in South Korean adolescents; (2) Methods: This population-based, cross-sectional study analyzed the data obtained from the Korea National Health and Nutrition Examination Survey between 2013 and 2015. Data related to 1422 adolescents aged 12–18 years (741 males and 681 females) were included in the analysis. Sleep duration was assessed using a self-reported questionnaire. Nutrient intake, including daily total energy intake, was assessed with a 24-h dietary recall questionnaire; (3) Results: Most males (84.4%) and females (86.4%) reported < 9 h of sleep per night. Short sleep duration was inversely associated with body mass index and obesity in both sexes. We found that higher intake of fiber and lower intake of sodium were associated with longer sleep duration (*P* < 0.05). When comparing the intake above and below the estimated average requirements (EAR), the difference in sleep duration was significant in the group that consumed vitamins B_1_ and C below EAR; (4) Conclusions: The findings of this study indicate that sleep duration can be associated with intake of some nutrients, which may also be associated with obesity in adolescents. Therefore, it is possible to prevent obesity and its complications by controlling the sleep duration and intake of nutrients of adolescents.

## 1. Introduction

Adolescence is a transient period from childhood to adulthood, which is characterized by rapid physical growth and psychological changes, including sleep. Sleep architecture changes throughout adolescence, showing a marked reduction in slow wave sleep (SWS) and rapid eye movement (REM) sleep in absolute terms, but not as a percentage of total sleep time [1]. Moreover, short sleep duration in adolescence is influenced by external factors such as use of electronic devices and social networking [2]. However, the sleep need does not change during adolescence—the estimated median sleep length necessary for adolescents to sustain waking vigilance and alertness is 9 h [3,4]. Insufficient sleep may interfere with the physiological restoration that occurs during this period, leading to biological and behavioral risk factors for chronic disease development including obesity, depression, and cardiovascular disease [5,6,7,8].

The rates of obesity among adolescents have increased over the past 3 decades; currently, more than 330 million children and adolescents worldwide were overweight or obese [9]. Obesity during adolescence is a predictor of obesity in adulthood [10]. Therefore, prevention efforts should begin early in life to prevent complication of obesity in adulthood. Since the relationship between insufficient sleep and obesity has been observed in both children and adults, the potential links between sleep, dietary intake, and nutrition have received increased attention. Although the exact mechanism of association between dietary intake and sleep duration is not clear, several possibilities have been suggested as follows: (1) short sleep duration increased free time to eat foods [11]; (2) sleep deprivation has been associated with activation of the stress system, which has been associated with decreased leptin and elevated ghrelin levels and increased insulin insensitivity [12,13], and a short sleep duration has been associated with decreased levels of thyroid stimulating hormone and growth hormone that stimulate the basal metabolic rate [14,15], whereas some studies have reported that manipulating the diet can alter sleep duration and quality. High intake of fish and vegetables was associated with good sleep quality [16]. The Mediterranean diet was associated with insomnia in older adults [17].

There have been studies on the association between dietary habits or food intake, such as intake of sweetened food or beverages, and sleep duration, but there are few studies on the association between nutrients intake and sleep duration in adolescents. Therefore, this study examined the association between sleep duration and dietary nutrients intake presented in the results of the 6th Korean Nutrition Health and Nutrition Examination Survey (KNHANES).

## 2. Materials and Methods

### 2.1. Survey Overview and Study Subjects 

This population-based, cross-sectional study analyzed the data of South Korean adolescents who participated in the Korea National Health and Nutrition Examination Survey (KNHANES) between 2013 and 2015. The KNHANES, a nationwide ongoing survey of non-institutionalized civilians, has been conducted by the Division of Chronic Disease Surveillance of the Korea Centers for Disease Control and Prevention (KCDC) and the Korean Ministry of Health and Welfare since 1998 [18]. The survey was designed to assess national health and nutritional status and consists of a health interview, nutritional assessment, and health examination. Participants were randomly selected from sampled household units using a stratified, multistage, and probability-based sampling design based on the component ratio of population and the housing census from the 2005 National Census Registry in South Korea. 

Of 22,948 participants sampled in the KNHANES 2013–2015, we excluded 21,093 individuals who were aged <12 years or >18 years and 433 individuals who had missing values for major target variables, such as nutritional survey and sleep duration, and anthropometric measures. A total of 1422 adolescents (741 males and 681 females) were analyzed. All study subjects were considered minors; therefore, their parents signed the written informed consent forms. The institutional review board of the KCDC approved the study protocol. The survey data are available from the KCDC website (http://knhanes.cdc.go.kr). 

### 2.2. Sleep Duration Measurements

Sleep duration was assessed using a self-reported questionnaire. All subjects were asked about their sleep duration, smoking status, alcohol consumption, physical activity, and monthly household income level. Sleep duration data was acquired from the following question in a self-reported questionnaire: “How many hours do you sleep on average?” Sleep duration was classified into five categories- ≤5 h, 6 h, 7 h, 8 h, and ≥9 h.

### 2.3. Lifestyle Variables and Nutritional Assessment 

Data on age, area of residence, education level, occupation, monthly household income level, physical activity, smoking status, and alcohol consumption were obtained during the health-related interview. We divided the subjects into non-smokers and current smokers according to their self-reported smoking status. Subjects who had smoked cigarettes at least once during the month before the survey were defined as current smokers. Regarding alcohol consumption, subjects who had at least one alcoholic drink once during the month before the survey were defined as alcohol drinkers. Physical activity was assessed using minutes of walking per day calculated from weekly frequency and minutes of walking. Monthly household income level was divided into the lower 25th percentile of the total subjects or higher. 

### 2.4. Anthropometry and Body Composition Measurements

Trained staff members performed anthropometric and biochemical measurements. Height and body weight (BWt) were measured to the nearest 0.1 cm and 0.1 kg, respectively, while the participants were barefoot and wearing light clothing. Waist circumference was measured at the narrowest point between the lower border of the rib cage and the iliac crest while the participants were in standing position. Body mass index (BMI) was calculated by dividing BWt (kg) by height (m) squared. 

BMI was categorized into 4 groups according to American Medical Association Expert Committee Recommendations [19]: underweight (BMI < 5th percentile), healthy weight (BMI between the 5th and 84th percentiles), overweight (BMI between the 85th and 94th percentiles), and obese (BMI ≥ 95th percentile). 

### 2.5. Nutrients Intake Asessment

Nutrient intake, including daily total energy intake, was assessed using a 24-h dietary recall questionnaire administered by a trained dietician. The 24-h recall method examines the type and intake of food consumed one day before the survey date using tools such as 2D images, measuring cups, and thick slice. Energy and nutrient intakes for each participant were calculated using the Korean Foods and Nutrients Database of the Rural Development Administration [20]. Food groups were analyzed into 18 categories based on the Standard Food Composition Table, 8th Revision in Korea [21]. The amount of nutrients intake was compared with the Dietary Reference Intakes for Koreans 2015 [22]. 

### 2.6. Statistical Analyses

Statistical analyses were performed with the statistical software package (SAS version 9.4 for Windows, SAS Institute, Cary, NC, USA) using SAS survey procedures including the appropriate weight, strata, domain, and cluster variables to account for the complex multistage sampling design in the KNHANES. Continuous variables (presented as mean ± standard error) were compared by sex using the independent t-test and by categories of sleep duration using the analysis of variance (ANOVA). To evaluate the appropriateness of nutrient intake, the intake of nutrients was divided into two groups (below the estimated average requirement (EAR) and above EAR) to test the association with sleep duration using the chi-square test [22]. 

Least square means of the sleep duration were compared according to BMI quartiles using the analysis of covariance (ANCOVA) after adjustment for age, smoking status, alcohol consumption, physical activity, monthly household income level, and energy intake (Figure 1).

ANCOVA was used with calculation of linear trend to evaluate the relationship between dietary nutrients and sleep duration while controlling for other potential confounders. Model 1 was unadjusted. In Model 2, adjustments were made for the variables age and sex. Model 3 was adjusted for the variables of Model 2, smoking status, alcohol consumption, physical activity, monthly household income level, and energy intake. *P* values were two tailed, and statistical significance was set at *P* < 0.05. 

## 3. Results

### 3.1. Demographic Characteristics of Study Participants

Baseline characteristics of the study population are presented in Table 1. The study participants consisted of 741 males and 681 females aged 12–18 (average, 15) years; 62.5% were aged 12–15 years and 37.5% were aged 16–18 years. Regarding BMI, 19.0% of participants were above the 85th percentile, and the average BMI was 21.7. 

Only 14.7% (n = 215) of participants reported more than 9 h sleep per night, with less than half of them (males, 42.8%; females, 33.0%) reporting below 8 h of sleep per night. Sleep duration differed significantly between sexes (*P* < 0.001). 

### 3.2. Sleep Duration and Obese Status

Figure 1 shows the sleep duration according to the quartiles of BMI. Sleep duration was inversely associated with BMI and obesity in both sexes (*P* < 0.05). 

### 3.3. Sleep Duration and Dietary Nutrients Assessments

There were no associations between sleep duration and total energy intake and intake of fat, carbohydrate, and protein (Table 2 and Table 3). After adjustment for confounding factors (age, smoking status, alcohol consumption, physical activity, monthly household income level, and energy intake), fiber and sodium intake were significantly associated with sleep duration. Higher intake of fiber and lower intake of sodium were associated with longer sleep time (*P* < 0.05, Table 3). When comparing the intake above and below EAR, the difference in sleep duration was significant in the group that consumed vitamins B1 and C below EAR (Table 3).

Model 1 was unadjusted. Model 2 was adjusted for age. Model 3 was adjusted for age, smoking, alcohol, physical activity, income and energy intake. 

## 4. Discussions 

The key finding of this national population-based study was the association between nutrients and sleep duration among adolescents. Particularly, shorter sleep duration was associated with decreased intake of fiber and increased intake of sodium. In addition, intake of vitamin B1 and C was associated with sleep duration. 

### 4.1. Sleep Duration Associated with Obesity 

In this study, we found an inverse association between sleep duration and obese status in South Korean adolescents, which is consistent with previous epidemiological studies performed in children and adolescents [23,24]. 

There are two most commonly reported associations between sleep duration and obesity. The first is an inverse linear correspondence where shorter sleep durations are associated with increased risk of obesity. This pattern suggests that more sleep is better for preventing obesity. In addition, a meta-analysis by Chen et al. has reported a linear dose–response association in children aged <10 years; however, studies in adolescents have inconsistent results [25]. In an 8-year prospective cohort study, shorter sleep duration was more strongly associated with a higher BMI in middle childhood than in adolescence [26]. 

The other pattern is a U-shaped association in which sleep durations that are longer or shorter than about 7–8 h/night were associated with increased risk of obesity [27,28]. 

The difference between these two sleep patterns may be related to whether the sleep needs are met according with age. Most adults need about 7–8 h of sleep per day, but infants and children need more. Physiologically, the younger the age, the longer the sleep need; children have a long sleep duration. However, adults have less sleep needs than children, but they often sleep less than they need to. Instead, long sleepers are often reported as mentally ill. Therefore, it is possible that insufficient sleep or excess sleep may explain the U-shaped association between sleep and obesity in older age. Future studies should investigate these differences by comparing different age groups.

### 4.2. Sleep Duration and Dietary Nutrients 

Our findings indicate that sleep duration may be associated with intake of fiber and vitamins, but not with fat, protein and carbohydrate It is still unclear whether sleep duration affects dietary intake or dietary intake affects sleep duration and quality. Some studies on the association between sleep duration and dietary nutrients have yielded different findings in adults. Changed dietary pattern after sleep deprivation is commonly associated with macronutrients such as fat or carbohydrates. In one study, the diets of adolescents aged 14–16 years after several nights of sleep restriction were characterized by higher glycemic index and glycemic load and a trend toward increased consumption of calories and carbohydrates [29]. In another study, older adolescents sleeping <8 h consumed a higher proportion of calories from fat and a lower proportion of calories from carbohydrates [30]. Moreover, pubertal insulin resistance is physiological, occurring with pubertal progression and resolving by the end of puberty, and is associated with decreased peripheral insulin sensitivity and increased insulin secretion [31,32]. Hyperinsulinemia is a marker of obesity and reduced SWS has been shown to influence insulin insensitivity in young adults [33]. Therefore, insufficient sleep may have a greater effect on obesity during adolescence than during childhood. 

Nutrients affect sleep architecture and pattern. A recent study reported that greater fiber intake predicted less sleep stage 1 and more SWS, and low fiber intake was associated with arousals in polysomnography of adults. [34]. Moreover, carbohydrates and proteins can influence the levels of neurotransmitters in the intrinsic sleep processes and affect sleep patterns. For example, short-term low proportion of carbohydrate intake was found to increase the percentage of SWS and reduce the percentage of REM sleep among healthy sleepers [35]. 

Vitamin C is an important antioxidant found in citrus fruits and vegetables. Low intake of vitamin C is associated with non-restorative sleep [36]. Vitamin B is involved in many metabolic functions, including sleep and circadian factors. Vitamin B has been shown to reduce daytime sleepiness and improve sleep pattern. A large study found that normal sleepers ingested significantly higher levels of vitamin B1 and vitamin B12 than those with insomnia [37]. Additionally, vitamin B1 deficiency has been associated with short sleep duration in adults, and adults who sleep <6 h have been shown to have a significantly lower intake of potassium, fiber, and calcium [38]. 

There have been studies supporting the association between fruit and vegetable intake and sleep dduration, with one indicating that adolescents who sleep >8 h consume more fruit and vegetables [39], and another showing that <7 h self-reported sleep is associated with reduced vegetable and fruit consumption compared with >8 h sleep duration [40]. Fruits and vegetables are rich in vitamins and dietary fiber. These studies support our findings and suggest that food choices may play a role in sleep duration in adolescents. 

Our study found an adverse association between sodium intake and sleep duration in adolescents. Sodium intake is associated with blood pressure, and elevated blood pressure is associated with short sleep duration and poor efficacy in adolescents and adults [41,42]. A national study has reported that salt is associated with sleep maintenance difficulties in adults (odds ratio = 1.19) [43]. The biological mechanisms underlying the association between short sleep duration and hypertension are unclear. Our finding regarding short sleep duration in a group with high intake of sodium may illustrate a possible link between insufficient sleep and hypertension. 

Diet could be useful in the management of sleep disturbances. It may be a self-feeding mechanism where dietary intake affects sleep pattern, which then affects food choices and leads to alterations in dietary consumption patterns.

### 4.3. Study Limitations and Strengths 

This study has several limitations. First, this study was based on KNHANES data and the associations do not necessarily imply causation. Our results suggest only exploratory evidence which is obtained without controlling false positive rates statistically. Therefore, future confirmatory studies are needed to find solid evidence of relationships. Second, like most previous studies, this study depended on self-reported sleep duration and there is no distinction between weekends and weekdays. Studies on the correlation between self-reported sleep duration and objectively measured sleep duration have reported moderate correlation [44,45]. Finally, the dietary data were collected using a 24-h dietary recall, limiting the ability to measure usual intake. However, under-reporting is less likely in 24-h dietary recalls than in self-reporting surveys in which participants are asked to record their own food intake [46]. In addition, at a population level, it can provide rich details about the mean dietary intake for a given day [47]

In conclusion, sleep duration can be associated with intake of some nutrients, which may also be associated with obesity in adolescents. Therefore, it is possible to prevent obesity and its complications by controlling the sleep duration and intake of nutrients of adolescents. 

## Figures and Tables

**Figure 1 children-07-00221-f001:**
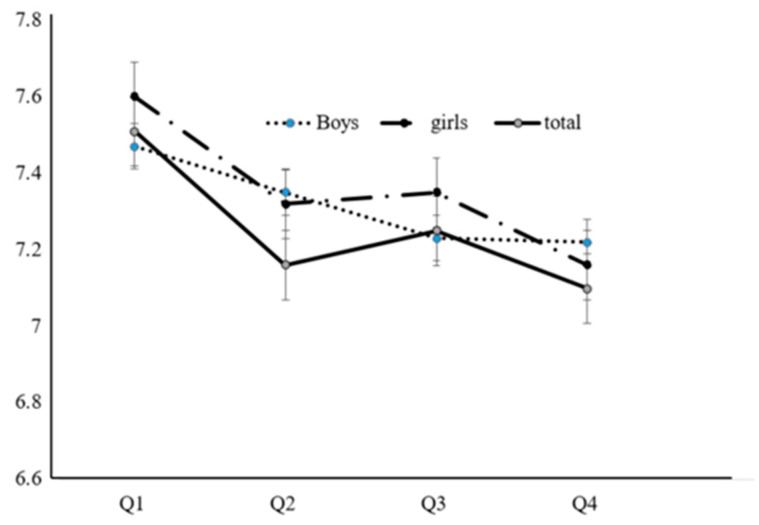
Adjusted sleep duration according to body mass index in both sexes. BMI: Body mass index. Adjusted for age, smoking, alcohol, physical activity, income, energy intake, each *P* value < 0.05. Q1: lowest BMI quartiles, Q2: middle–low BMI quartiles, Q3 middle–high BMI quartiles, Q4: highest BMI quartiles.

**Table 1 children-07-00221-t001:** Baseline characteristics of study subjects.

Study Population (n = 1422)	Male (n = 741)	Female (n = 681)	*P*
Age(years)	15.1 ± 0.07	15.1 ± 0.08	0.747
Height (cm)	169.3 ± 0.36	159.8 ± 0.27	<0.001 *
Weight (kg)	62.6 ± 0.58	54.3 ± 0.47	<0.001 *
Waist Circumference (cm)	74.0 ± 0.43	69.3 ± 0.36	<0.001 *
BMI (kg/m^2^)	21.7 ± 0.18	21.2 ± 0.15	0.020 *
Energy intake (cal)	2567.2 ± 50.7	1890.5 ± 33.4	<0.001 *
Fat intake (%)	24.9 ± 0.4	24.0 ± 0.4	0.085
Walking activity (per day)	33.8 ± 1.68	31.0 ± 1.49	0.195
Sleep Duration (hours, %)			<0.001 *
<6 h	9.7	16.7	
6 to <8 h	47.5	50.3	
8 to <9 h	27.2	19.4	
≥9 h	15.6	13.6	

Values are presented as the mean ± standard error or percentage, * *P* < 0.05.

**Table 2 children-07-00221-t002:** Comparison between amount of dietary nutrient intake and sleep duration.

Sleep Duration				<6 h	6 to <8 h	8 to <9 h	≥9 h	*P*
N (%)	Intake	n	%	157 (13.1)	690 (48.8)	360 (23.4)	215 (14.7)	
Carbohydrate (%)	≤65%	912	64.1	13.3	49.2	22.8	14.8	0.921
	>65%	510	35.9	12.8	48.1	24.5	14.6	
Fat (%)	≤30%	1082	76.1	13.3	49.3	24.1	13.3	0.162
	>30%	340	23.9	12.3	47.4	21.3	18.9	
Saturated fat (%)	<8%	814	57.2	14.3	48.5	23.9	13.3	0.273
	≥8%	608	42.8	11.5	49.3	22.7	16.6	
Protein (%)	<EAR	223	15.7	12.8	52.8	20.3	14.0	0.615
	≥EAR	1199	84.3	13.1	48.0	24.0	14.8	
Fiber (%)	<AI	1030	72.4	13.7	48.7	24.5	13.1	0.069
	≥AI	392	27.6	11.6	49.1	20.5	18.8	
Vitamin A (%)	<EAR	781	54.9	13.8	47.4	23.9	14.9	0.727
	≥EAR	641	45.1	12.2	50.7	22.7	14.4	
Vitamin C (%)	<EAR	966	67.9	14.9	47.9	23.1	14.1	0.040 *
	≥EAR	456	32.1	9.1	50.9	24.0	16.0	
Sodium (%)	<AI	144	10.1	12.9	51.4	22.7	13.1	0.927
	≥AI	1278	89.9	13.1	48.5	23.5	14.9	
Potassium (%)	<AI	1116	78.5	13.7	48.1	24.3	13.9	0.158
	≥AI	306	21.5	10.7	51.7	20.0	17.6	
Calcium (%)	<EAR	1177	82.8	13.4	48.7	24.0	13.9	0.276
	≥EAR	245	17.2	11.6	49.6	20.4	18.4	
Phosphorus (%)	<EAR	667	46.9	15.2	49.3	21.7	13.7	0.127
	≥EAR	755	53.1	11.1	48.4	25.0	15.6	
Iron (%)	<EAR	548	38.5	15.7	46.0	24.0	14.3	0.161
	≥EAR	874	61.5	11.4	50.6	23.0	14.9	
VitaminB1 (%)	<EAR	145	10.2	18.4	51.8	22.7	7.1	0.035 *
	≥EAR	1277	89.8	12.4	48.5	23.5	15.6	
Vitamin B2 (%)	<EAR	545	38.3	15.6	47.3	24.7	12.4	0.074
	≥EAR	877	61.7	11.5	49.8	22.5	16.2	
Niacin (%)	<EAR	464	32.6	15.6	48.8	23.8	11.8	0.109
	≥EAR	958	67.4	11.8	48.8	23.2	16.2	

Values are presented as the mean ± standard error or percentage. AI, Adequate Intake; EAR, Estimated Average Requirements; * *P* < 0.05.

**Table 3 children-07-00221-t003:** Association between sleep duration and dietary nutrient intake.

Total N		Total n = 1422				
Sleep Duration (h/night)		<6 h (n = 157)	6 to <8 h (n = 690)	8 to <9 h (n = 362)	≥9 h (n = 216)	*p* for Trends
Nutrients (Unit)		Mean	Mean	Mean	Mean	
Carbohydrate (%)	model 1	61.1 ± 0.9	61.1 ± 0.4	61.5 ± 0.7	60.0 ± 0.8	0.446
	model 2	61.8 ± 0.9	61.5 ± 0.4	61.4 ± 0.7	59.8 ± 0.8	0.137
	model 3	61.1 ± 1.1	61.1 ± 0.9	61.0 ± 0.9	59.6 ± 1.1	0.221
Protein (%)	model 1	14.8 ± 0.4	14.4 ± 0.2	14.4 ± 0.3	14.3 ±0.3	0.282
	model 2	14.6 ± 0.4	14.3 ± 0.2	14.5 ± 0.3	14.3 ± 0.3	0.752
	model 3	14.4 ± 0.5	14.1 ± 0.4	14.3 ± 0.4	14.0 ± 0.5	0.558
Fat (%)	model 1	24.1 ± 0.8	24.5 ± 0.3	24.1 ± 0.5	25.7 ± 0.7	0.185
	model 2	23.7 ± 0.8	24.2 ± 0.3	24.1 ± 0.6	25.8 ± 0.7	0.084
	model 3	24.4±1.0	24.8 ± 0.7	24.8 ± 0.8	26.3 ± 1.0	0.109
Carbohydrate (g/day)	model 1	305.9 ± 13.2	335.9 ± 6.2	328.8 ± 7.5	333.6 ± 11.2	0.152
	model 2	319.7 ± 13.3	337.2 ± 5.7	319.3 ± 7.1	328.0 ± 11.6	0.900
	model 3	322.6 ± 7.3	324.7 ± 6.0	316.2 ± 7.6	320.4±6.5	0.553
Protein (g/day)	model 1	75.9 ± 4.9	82.1 ± 2.2	78.6 ± 2.6	79.6 ± 3.1	0.660
	model 2	77.1 ± 4.8	81.2 ± 2.0	76.3 ± 2.5	78.9 ± 3.3	0.976
	model 3	79.2 ± 3.3	78.4 ± 2.4	76.4 ± 2.8	76.9 ± 2.9	0.367
Fat (g/day)	model 1	57.7 ± 4.6	62.9 ± 1.8	60.1 ± 2.5	64.3 ± 3.2	0.328
	model 2	58.2 ± 4.7	62.1	58.5 ± 2.4	63.8 ± 3.4	0.496
	model 3	62.5 ±	62.4	61.0 ± 2.1	64.7 ± 2.3	0.547
Saturated fat (g/day)	model 1	18.0 ± 1.4	20.0 ± 0.6	19.4 ± 0.8	20.7 ± 1.2	0.164
	model 2	18.5 ± 1.4	19.8 ± 0.5	18.8 ± 0.8	20.5 ± 1.2	0.398
	model 3	19.0 ± 0.9	19.1 ± 0.7	18.8 ± 0.8	20.0 ± 1.1	0.463
Cholesterol (mg/d)	model 1	273.5 ± 21.0	326.7 ± 11.3	299.6 ± 14.0	324.3±24.1	0.184
	model 2	271.8 ± 22.5	322.3 ± 10.7	294.2 ± 14.0	323.8 ± 24.4	0.214
	model 3	263.8 ± 19.1	293.0 ± 16.6	275.2 ± 17.7	297.7 ± 24.6	0.307
Fiber (g/day)	model 1	16.7 ± 0.9	18.5 ± 0.4	18.4 ± 0.6	20.6 ± 1.2	0.008 *
	model 2	17.1 ± 0.9	18.5 ± 0.4	18.0 ± 0.6	20.4 ± 1.2	0.039 *
	model 3	16.5 ± 0.9	17.0 ± 0.7	17.0 ± 0.8	19.3 ± 1.0	0.022 *
Vitamin A (mcg/d)	model 1	580.2 ± 62.0	670.7 ± 33.4	680.2 ± 54.5	730.4 ± 134.4	0.307
	model 2	630.3 ± 70.9	682.4 ± 35.4	656.7 ± 55.0	712.3 ± 129.0	0.584
	model 3	579.5 ± 71.6	593.9 ± 47.1	576.2 ± 73.1	628.2 ± 117.0	0.748
Vitamin C (mg/d)	model 1	59.7 ± 6.6	77.4 ± 4.6	74.2 ± 5.1	77.1 ± 6.1	0.078
	model 2	63.9 ± 6.7	78.5 ± 4.3	72.4 ± 5.3	75.6 ± 6.0	0.311
	model 3	61.0 ± 9.6	73.8 ± 11.2	70.1 ± 10.7	73.4 ± 10.3	0.248
Sodium (mg/d)	model 1	3704.1 ± 245.9	3701.6 ± 106.7	3431.3 ± 109.2	3429.2 ± 148.5	0.170
	model 2	3734.7 ± 238.4	3658.2 ± 98.0	3338.5 ± 103.6	3403.7 ± 156.9	0.112
	model 3	3890.9 ± 172.4	3638.4 ± 126.1	3436.6 ± 139.9	3433.2 ± 161.1	0.005 *
Potassium (mg/d)	model 1	2401.2 ± 120.1	2715.4 ± 57.8	2597.2 ± 77.8	2800.8 ± 128.7	0.039
	model 2	2487.0 ± 124.2	2716.5 ± 53.1	2527.6 ± 75.0	2764.4 ± 132.0	0.243
	model 3	2446.4 ± 100.1	2532.1 ± 75.8	2410.7 ± 80.8	2607.5 ± 107.4	0.413
Calcium (mg/d)	model 1	434.1 ± 24.4	483.5 ± 10.8	489.1 ± 18.0	555.3 ± 47.4	0.015
	model 2	454.5 ± 25.9	486.5 ± 10.6	476.7 ± 17.7	547.4 ± 48.7	0.127
	model 3	437.8 ± 27.5	448.7 ± 22.5	451.8 ± 25.4	521.3 ± 55.1	0.129
Phosphorus (mg/d)	model 1	1015.4 ± 49.0	1136.5 ± 24.1	1114.8 ± 32.7	1154.8 ± 42.1	0.045
	model 2	1052.8 ± 51.5	1136.1 ± 22.1	1083.3 ± 31.7	1138.7 ± 44.3	0.357
	model 3	1033.0 ± 28.6	1052.0 ± 23.4	1031.7 ± 26.4	1068.4 ± 32.6	0.448
Iron (mg/d)	model 1	14.9 ± 1.1	16.6 ± 0.6	18.4 ± 2.7	15.4 ± 0.7	0.506
	model 2	15.5 ± 1.4	16.6 ± 0.6	17.9 ± 2.5	15.2 ± 0.7	0.947
	model 3	15.8 ± 1.9	16.0 ± 1.1	17.8 ± 2.6	14.9 ± 1.2	0.822
Vitamin B1(mg/d)	model 1	1.96 ± 0.13	2.14 ± 0.05	2.00 ± 0.06	2.08 ± 0.08	0.654
	model 2	1.98 ± 0.13	2.11 ± 0.05	1.95 ± 0.06	2.06 ± 0.08	0.882
	model 3	2.03 ± 0.12	2.06 ± 0.11	1.96 ± 0.10	2.03 ± 0.10	0.662
Vitamin B6 (mg/d)	model 1	1.32 ± 0.07	1.50 ± 0.03	1.46 ± 0.06	1.51 ± 0.06	0.057
	model 2	1.35 ± 0.08	1.50 ± 0.03	1.42 ± 0.06	1.49 ± 0.06	0.306
	model 3	1.36 ± 0.09	1.42 ± 0.08	1.38 ± 0.08	1.42 ± 0.08	0.501
Niacin (mcg/d)	model 1	15.99 ± 1.19	16.75 ± 0.47	16.28 ± 0.73	16.40 ± 0.64	0.853
	model 2	16.13 ± 1.17	16.55 ± 0.45	15.85 ± 0.71	16.29 ± 0.67	0.960
	model 3	16.60 ± 0.91	15.97 ± 0.63	15.84 ± 0.62	15.85 ± 0.72	0.347

Values are presented as the mean ± standard error or percentage, * *P* < 0.01.

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
