# Peer review of "Association between Self-Reported Sleep Duration and Dietary Nutrients in Korean Adolescents: A Population-Based Study"

_children, 2020, doi:10.3390/children7110221_

Round 1

Reviewer 1 Report

Thank you for the opportunity to review the paper. Overall, the paper is well-written, though I have some reservations about the premise and approach. I have grouped my comments into major and minor points below:

Major

  1. The premise of the study is somewhat unclear. The authors cite possible mechanisms for which sleep may impact weight, but these studies (e.g. 10,11) despite being over-cited, have weak scientific premise (e.g. small sample size, lack of robust methodologies etc.). Significant revisions to the introduction to explain the potential association between sleep and diet is needed. More references (including those that are published within the past few years) are needed.
  2. ANOVA also relies heavily on normality assumptions. Briefly discuss if these assumptions were violated, and provide a statement on effect size and power calculations. For large sample sizes like this, I am surprised to see ANOVA being used instead of (Ordinal) Logistic regression or similar.
  3. A major issue I have is with the problem of multiple comparisons. Taking a look at the large number of variables presented in Tables 2 and 3, I am surprised that adjustments (e.g. Bonferroni correction) were not used. The significant p-values (i.e. .040,.035,.022) are dangerously close to the cutoff of .05, and it is clear that once something like Bonferroni correction is applied, these p values will no longer be significant. The authors must very clearly justify why steps to address multiple comparisons were not taken. Otherwise, analysis may need to be repeated or overhauled.

Minor points

  1. Since 433 individuals were excluded, which is a large chunk of participants overall, briefly describe their characteristics and if it differs from the main sample size.
  2. I am unsure if physical activity can be calculated in this manner. PA is multi-faceted and includes more than just walking. Particularly among adolescents who engage in greater level vigorous activity, walking may not be an appropriate gauge of PA for this age group. Please amend or justify by including appropriate references.
  3. Throughout the methods section, provide references.
  4. Table 1: Consider presenting (or even analyzing) sleep duration as a continuous variable. Analyzing as a continuous variable will allow you to preserve power, unless the variable plot is non-linear.
  5. Include error bars in figure 1.
  6. Move the first two paragraphs in "Sleep need of adolescents" subsection of the discussion section into the introduction section. Discussion sections should be strictly limited to interpreting and discussing the results of the study, not introduce concepts in the literature. 
  7. In the limitations section, the paper (44) cited to explain how self-reported sleep is correlated with objective measurements is inaccurate. The reference does not examine correlations, only that there are no significant differences between child self-report and objective measurements (two different statistical tests). Please update or rephrase.
  8. Be careful not to say that "Our findings suggest that high-fiber...may IMPROVE sleep duration". Using the word "improve" suggests causality/directionality.

Author Response

Major

  1. The premise of the study is somewhat unclear. The authors cite possible mechanisms for which sleep may impact weight, but these studies (e.g. 10,11) despite being over-cited, have weak scientific premise (e.g. small sample size, lack of robust methodologies etc.). Significant revisions to the introduction to explain the potential association between sleep and diet is needed. More references (including those that are published within the past few years) are needed.

[Response] Thank you for your careful review.

We have revised many parts of the manuscript, especially the introduction and conclusion as your comment. We tried to explain potential association between sleep and diet. In addition, some references have been removed and replaced with other references that are published within past few years.

Original) page 1~2

  1. Introduction

Adolescence is an important developmental stage characterized by rapid spurt in physical growth and psychological changes. Regarding sleep, adolescents have a later bedtime and shorter sleep duration partly because of pubertal changes in the homeostatic and circadian regulations of sleep[1]. Moreover, short sleep duration in adolescence is influenced by external factors such as use of electronic devices and social networking [2].  

Insufficient sleep may interfere with the physiological restoration that occurs during this period, leading to biological and behavioral risk factors for chronic disease development including obesity, depression, and cardiovascular disease [3-5].  

The rates of obesity among adolescents have increased over the past 3 decades; currently, more than one-third of children and adolescents worldwide are obesity[6]. Since the relationship between insufficient sleep and obesity has been observed in both children and adults, the potential links between sleep, dietary intake, and nutrition have received increased attention. However, the mechanism of association between dietary nutrients and sleep duration is not well known. Some studies have found that insufficient sleep can affect dietary choice and hunger, whereas others have found that manipulating diet can alter sleep architecture and sleep patterns [7-10].  

The mechanism by which sleep deprivation may affect weight is unknown, but several possibilities have been suggested such as: 1) increased free time to eat foods; 2) sleep deprivation has been associated with activation of the stress system, which has been associated with decreased leptin and elevated ghrelin levels and increased insulin insensitivity [10, 11], and a short sleep duration has been associated with decreased levels of thyroid stimulating hormone and growth hormone that stimulate basal metabolic rate [11, 12]; and 3) tiredness due to sleep deprivation could lead to a decrease in physical activity and consequent reduction of energy expenditure [13].  

Dietary patterns established during adolescence tend to persist into adulthood. In addition, pediatric sleep disturbance can continue into adulthood. Therefore, healthy eating and sleeping habits are important for all stages of life. However, the causality or the direction of the relation between sleep and obesity is not well known. It can be hypothesized that sufficient sleep can modify eating behavior; thus, promoting healthy sleep habits is a good strategy for preventing obesity in adolescents. 

To our knowledge, this is the first study on the association between sleep duration and dietary nutrients using population-based data. In addition, few studies have compared the association between dietary nutrients intake and sleep duration in adolescents with that in adults and young children. Therefore, this study aimed to examine the association between sleep duration and dietary nutrients intake in South Korean adolescents. 

Revision) page 1~2

Adolescence is a transient period from childhood to adulthood, which characterized by rapid physical growth and psychological changes, including sleep Sleep architecture changes throughout adolescence, showing marked reduction in slow wave sleep (SWS) and rapid eye movement (REM) sleep decreases in absolute aspect, but not as a percentage of total sleep time [1]. In addition, short sleep duration in adolescence is influenced by external factors such as use of electronic devices and social networking [2]. However, the sleep need does not change during adolescence—the estimated median sleep length necessary for adolescents to sustain waking vigilance and alertness is 9 h [3, 4].  Insufficient sleep may interfere with the physiological restoration that occurs during this period, leading to biological and behavioral risk factors for chronic disease development including obesity, depression, and cardiovascular disease [5-8]

The rates of obesity among adolescents have increased over the past 3 decades; currently, more than 330 million children and adolescents worldwide were overweight or obese [9]. Obesity during adolescence is a predictor of obesity in adulthood[10]. Therefore, prevention efforts should begin early in life to prevent complication of obesity in adulthood. Since the relationship between insufficient sleep and obesity has been observed in both children and adults, the potential links between sleep, dietary intake, and nutrition have received increased attention. Although, the exact mechanism of association between dietary intake and sleep duration is not clear, several possibilities has been suggested as follows; 1) short sleep duration increased free time to eat foods[11]; 2) sleep deprivation has been associated with activation of the stress system, which has been associated with decreased leptin and elevated ghrelin levels and increased insulin insensitivity[12, 13], and a short sleep duration has been associated with decreased levels of thyroid stimulating hormone and growth hormone that stimulate basal metabolic rate [14, 15]

whereas some studies have reported that manipulating diet can alter sleep duration and quality. High intake of fish and vegetables was associated with good sleep quality [16]. The Mediterranean diet was associated with insomnia in older adults.[17]

There have been studies on the association between dietary habits or food intake such as sweeten or beverage and sleep duration, but there are few studies for association between nutrients intake and sleep duration in adolescents. Therefore, this study examined the association between sleep duration and dietary nutrients intake presented in the results of the 6TH Korean Nutrition Health and Nutrition Examination Survey (KNHANES).

  1. ANOVA also relies heavily on normality assumptions. Briefly discuss if these assumptions were violated, and provide a statement on effect size and power calculations. For large sample sizes like this, I am surprised to see ANOVA being used instead of (Ordinal) Logistic regression or similar.

[Response]

We understand that it is important to check normality assumption. We tested normality assumption based on the Shapiro-Wilks test. We judged that normality assumption was not satisfied if the p-value of the Shapiro-Wilks test is less than 0.05. The p-value of the Shapiro-Wilks test is less than 0.05 in several nutrients. In those cases, we tried the square root transformation and the logarithm transformation. Unfortunately, the p-value of the Shapiro-Wilks test is still less than 0.05 in those cases. Therefore, our next option was a nonparametric approach that does not depend on normality assumption. However, PROC SURVEY (considering stratified, multistage sampling design of KNHANES) in SAS do not provide nonparametric ANOVA. Therefore, we ran out of options. Please understand this situation and consider the fact that almost all statistical analysis can not meet all required assumptions. It is important to figure out effect size and power calculations. Unfortunately, it is our understanding that PROC SURVEY in SAS does not provide a tool to do it.

In Table 3, all response variables are continuous. Therefore, we believe that ANOVA is appropriate. In Table 2, all response variables are binary. In such cases, it is known that the chi-square test and the logistic regression are asymptotically equivalent

  1. A major issue I have is with the problem of multiple comparisons. Taking a look at the large number of variables presented in Tables 2 and 3, I am surprised that adjustments (e.g. Bonferroni correction) were not used. The significant p-values (i.e. .040,.035,.022) are dangerously close to the cutoff of .05, and it is clear that once something like Bonferroni correction is applied, these p values will no longer be significant. The authors must very clearly justify why steps to address multiple comparisons were not taken. Otherwise, analysis may need to be repeated or overhauled.

[Response]

Multiple comparisons are important and controversial issues. We partially agree that we should consider multiple comparisons. However, we believe that we need to distinguish confirmatory evidences and exploratory evidences. Multiple comparisons should be employed in order to avoid false positive results for confirmatory evidences as you suggested. However, we tried to find exploratory evidences in Table 2 and 3 for this study. We think that exploratory evidences are also important in some aspects. We cannot find confirmatory evidences at one time. Instead, we find exploratory evidences first. Then such exploratory evidences can be basis of future confirmatory studies.

Minor points

  1. Since 433 individuals were excluded, which is a large chunk of participants overall, briefly describe their characteristics and if it differs from the main sample size.

[Response]

We agreed with your comment and we add brief description about missing data.

And in the part of the analysis of 433 exclusions is possible, there was no significant difference including age, BMI, waist circumference, and nutrient intake. 

Original) Page 2

433 individuals who had missing data

Revision) Page 3

433 individuals who had missing values for major target variables, such as nutritional survey and sleep duration, and anthropometric measures.

  1. I am unsure if physical activity can be calculated in this manner. PA is multi-faceted and includes more than just walking. Particularly among adolescents who engage in greater level vigorous activity, walking may not be an appropriate gauge of PA for this age group. Please amend or justify by including appropriate references.

[Response] We agreed on the reviewer’s comments that walking alone was not enough to explain the whole physical activity level in children and adolescents.

The KNHANES was conducted to assess the physical activity level as defined by the International Physical Activity Questionnaire-short form (IPAQ-SF). The IPAQ-SF is used to estimate the overall PA level of an individual in metabolic equivalent (MET)-min/week by determining the duration (in minutes) and number of days (in 1 week) of engagement in three specific types of activity (walking, moderate-intensity activities, and high intensity activities) across a comprehensive set of domains (leisure time, work-related and transport-related physical activities and domestic and gardening activities) in the past 7 days.  However, in our data, the most common item was walking because there were many missing data in other activity data for physical activity. Therefore, we assumed that walking variable were able to reflect the commuting physical activity. The walking variable reflecting commuting physical level was analyzed as one of the adjusting variable of lifestyle variables for nutritional assessment

In this revision, we modified the physical activity to walking activity.

   Original) Page 4, Table 1

Table 1. Baseline characteristics of study subjects

Study population (n=1422)

Male (n=741)

Female (n=681)

P

Age(years)

15.1±0.07

15.1±0.08

0.747

Height (cm)

169.3±0.36

159.8±0.27

<0.001*

Weight (kg)

62.6±0.58

54.3±0.47

<0.001*

WC (cm)

74.0±0.43

69.3±0.36

<0.001*

BMI (kg/m2)

21.7±0.18

21.2±0.15

0.020*

Energy intake (cal)

2567.2±50.7

1890.5±33.4

<0.001*

Fat intake (%)

24.9±0.4

24.0±0.4

0.085

minutes of walking(per day)

33.8±1.68

31.0±1.49

0.195

Sleep Duration (hours,%)

<0.001*

≤5h

9.7

16.7

6 to 8<h

47.5

50.3

8 to <9h

27.2

19.4

≥9h

15.6

13.6

Revision) Page 4

Table 1. Baseline characteristics of study subjects

Study population (n=1422)

Male (n=741)

Female (n=681)

P

Age(years)

15.1±0.07

15.1±0.08

0.747

Height (cm)

169.3±0.36

159.8±0.27

<0.001*

Weight (kg)

62.6±0.58

54.3±0.47

<0.001*

WC (cm)

74.0±0.43

69.3±0.36

<0.001*

BMI (kg/m2)

21.7±0.18

21.2±0.15

0.020*

Energy intake (Cal)

2567.2±50.7

1890.5±33.4

<0.001*

Fat intake (%)

24.9±0.4

24.0±0.4

0.085

Walking activity (min/per day)

33.8±1.68

31.0±1.49

0.195

Sleep Duration (hours,%)

<0.001*

≤5h

9.7

16.7

6 to 8<h

47.5

50.3

8 to <9h

27.2

19.4

≥9h

15.6

13.6

Values are presented as the mean ± standard error or percentage, * P < 0.05

  1. Throughout the methods section, provide references.

[Response] We revised as your comment.

Three references were added and nutritional assessment section was described in more detail.

Original) Page 3

Nutrient intake, including daily total energy intake, was assessed using a 24-h dietary recall questionnaire administered by a trained dietician. The participants provided data on their intake of foods, including type and amount, during the past 24 h (midnight to midnight). The results were calculated using the food composition table developed by the National Rural Living Science Institute under the Rural Development Administration. 

Revision) Page 3

The 24-hour recall method to examine the type and intake of food consumed one day before the survey date using tools such as 2D images, measuring cups, and thick slice.  Energy and nutrient intakes for each participant were calculated using the Korean Foods and Nutrients Database of the Rural Development Administration [20].

Food groups were analyzed into 18 categories based on the Standard Food Composition Table, 8th Revision in Korea[21]. The amount of nutrients intake was compared with the Dietary Reference Intakes for Koreans 2015 [22].

  1. Table 1: Consider presenting (or even analyzing) sleep duration as a continuous variable. Analyzing as a continuous variable will allow you to preserve power, unless the variable plot is non-linear.

[Response] I agree with your comment. However, since this study is a study on association between sleep duration and nutrient intake, similar other previous studies have been studied as categorical variables, it was chosen because it was easy to compare with results of previous studies.

We are sure that it will be reflected in our next study.

  1. Include error bars in figure 1.

[Response] We made a new figure with error bars.

Original) Page 5 

Revision) Figure 1

Figure 1. Adjusted sleep duration according to body mass index in both sex

BMI: Body mass index. Adjusted for age, smoking, alcohol, physical activity, income, energy intake, each P value < 0.05.  Q1: lowest BMI quartiles, Q2: middle low BMI quartiles, Q3 middle high BMI quartiles, Q4: highest BMI quartiles

  1. Move the first two paragraphs in "Sleep need of adolescents" subsection of the discussion section into the introduction section. Discussion sections should be strictly limited to interpreting and discussing the results of the study, not introduce concepts in the literature. 

[Response] We agree with your comments. Most of introduction section is revised.

Two paragraphs in the discussion section you comment have been moved to the introduction, and the introduction has been largely revised. 

Original) Page 8

Sleep need of adolescents

Most South Korean adolescents in this study slept <9 h. Adolescence is a transient period from childhood to adulthood. Sleep architecture changes throughout adolescence, showing marked reduction in slow wave sleep (SWS). In addition, rapid eye movement (REM) sleep decreases in absolute aspect, but not as a percentage of total sleep time [1]. However, the sleep need does not change during adolescence—the estimated median sleep length necessary for adolescents to sustain waking vigilance and alertness is 9 h [17, 18].

The short sleep duration among adolescence has been attributed to delay of circadian phases, attenuation of the build-up of sleep pressure, and external environment factors [19, 20]. Two primary hypotheses have been proposed to explain the puberty-related circadian delay of adolescents. One postulates that endogenous circadian periods may expand over the course of puberty, while the second hypothesis suggests an altered circadian response to light. The external environment factors that influence sleep patterns in adolescents and contribute to the delay in sleep timing and insufficient sleep include school start times, socioeconomic status, digital media use, social engagement, and caffeine intake. In addition, a competitive social atmosphere that values high-level education reduces sleep duration for adolescents [21]. These external factors are comparatively easy to control. The delay of school start time has been associated with self-reported school performance improvement and reduced duration of recovery sleep in the weekend, with a transient increase in sleep duration in South Korean adolescents [22].

       Reivision)

Most of removed sentences into introduction. The revised introduction was in response of major 1.

  1. In the limitations section, the paper (44) cited to explain how self-reported sleep is correlated with objective measurements is inaccurate. The reference does not examine correlations, only that there are no significant differences between child self-report and objective measurements (two different statistical tests). Please update or rephrase.

[Response] We agreed with your comment. And we add the references and revision of sentence in the limitation section. 

Original) Page 10

 Second, like most previous studies, this study depended on self-reported sleep duration and there is no distinction between weekends and weekdays. However, self-reported sleep duration is highly correlated with objective measurements in adolescents [44].

         Revision) Page 10

Second, like most previous studies, this study depended on self-reported sleep duration and there is no distinction between weekends and weekdays. Studies on the correlation between self-reported sleep duration and objectively measured sleep duration have reported moderate correlation[43, 44].

  1. Be careful not to say that "Our findings suggest that high-fiber...may IMPROVE sleep duration". Using the word "improve" suggests causality/directionality

[Response] We agree with your comment. We revised conclusion in abstract and discussion section.

Original) Page 2

Page 1- introduction

Sleep duration was associated with nutrients in adolescents. Our findings suggest that high-fiber, low-sodium and vitamins diets may improve sleep duration in adolescents with insufficient sleep. It is necessary to take more interest in the sleep duration and intake of dietary nutrients of adolescents and make more efforts toward preventing obesity.

Revision)

 Page 1- abstract / Page 2- Introduction / Page 10

     The findings of this study indicate that sleep duration can be associated with intake of some nutrients, which may also be associated to obesity in adolescents. Therefore, it is possible to prevent obesity and its complication by controlling sleep duration and intake of nutrients of adolescents

Reviewer 2 Report

The authors examined possible associations between sleep duration, nutritients' intake and obesity in adolescents of South Korea. It is an interesting study. Epidemiological data in this age are generally missing. Moreover, food intake, sleep and obesity are very closely linked.

Disadvantages of the study is the lack of information about daily quantity of food intake (kcals/24h) and the subjective evaluation of sleep duration.

Author Response

Response to Reviewer 2 Comments

The authors examined possible associations between sleep duration, nutrient intake and obesity in adolescents of South Korea. It is an interesting study. Epidemiological data in this age are generally missing. Moreover, food intake, sleep and obesity are very closely linked.

Disadvantages of the study is the lack of information about daily quantity of food intake (kcals/24h) and the subjective evaluation of sleep duration.

[Response] Thank you for your comments and we understand your comments for disadvantage.  We have added the disadvantage you mentioned to the limitation section. Tthe references and revision of sentence were revised.  

Original) Page 10

This study has several limitations. First, this study was based on KNHANES data and the associations do not necessarily imply causation. Second, like most previous studies, this study depended on self-reported sleep duration and there is no distinction between weekends and weekdays. However, self-reported sleep duration is highly correlated with objective measurements in adolescents [44]. Finally, the dietary data were collected using a 24-h dietary recall, limiting the ability to measure usual intake. However, under-reporting is less likely in 24-h dietary recalls than in self-reporting surveys in which participants are asked to record their own food intake [45].

Revision) Page 10

This study has several limitations. First, this study was based on KNHANES data and the associations do not necessarily imply causation. Second, like most previous studies, this study depended on self-reported sleep duration and there is no distinction between weekends and weekdays. Studies on the correlation between self-reported sleep duration and objectively measured sleep duration have reported moderate correlation[44, 45]. Finally, the dietary data were collected using a 24-h dietary recall, limiting the ability to measure usual intake. However, under-reporting is less likely in 24-h dietary recalls than in self-reporting surveys in which participants are asked to record their own food intake [46]. In addition, a population level, it can provide rich details about mean dietary intake for a given day [47]
